# Model and Feature Diversity for Bayesian Neural Networks in Mutual Learning

**Cuong Pham**[1]     **Cuong C. Nguyen**[2]     **Trung Le**[1]     **Dinh Phung** [1,4]

**Gustavo Carneiro**[3]     **Thanh-Toan Do**[1]

[1]Department of Data Science and AI, Monash University, Australia

[2]Australian Institute for Machine Learning, University of Adelaide, Australia

[3]Centre for Vision, Speech and Signal Processing, University of Surrey, United Kingdom

[4]VinAI, Vietnam

{cuong.pham1, trunglm, dinh.phung, toan.do}@monash.edu

cuong.nguyen@adelaide.edu.au, g.carneiro@surrey.ac.uk

## Abstract

Bayesian Neural Networks (BNNs) offer probability distributions for model parameters, enabling uncertainty quantification in predictions. However, they often underperform compared to deterministic neural networks. Utilizing mutual learning can effectively enhance the performance of peer BNNs. In this paper, we propose a novel approach to improve BNNs performance through deep mutual learning. The proposed approaches aim to increase diversity in both network parameter distributions and feature distributions, promoting peer networks to acquire distinct features that capture different characteristics of the input, which enhances the effectiveness of mutual learning. Experimental results demonstrate significant improvements in the classification accuracy, negative log-likelihood, and expected calibration error when compared to traditional mutual learning for BNNs.

## 1 Introduction

Bayesian neural networks [22] (BNNs) provide a probability distribution over the model parameters, which allows for uncertainty quantification in the model's predictions. This is especially useful in situations where making accurate predictions is crucial, and having a measure of uncertainty can inform decision-making. This measurement is essential for many applications, including continual learning, active learning, and robust operation. To measure uncertainty, we can learn the posterior distribution after observing data through methods like Variational Inference (VI) [1, 16] and Markov Chain Monte Carlo (MCMC) [37, 21, 3]. The variational approach for BNNs involves selecting prior distributions and approximate posterior distributions over neural network weights, offering a more memory-efficient and flexible approach compared to MCMC.

However, despite their advantages, Bayesian neural networks using variational inference often find it difficult to match the performance of BNNs using the MCMC approach, as well as deterministic neural networks. To address this performance gap, various methods have been explored [26, 28, 31, 8, 24, 14]. One promising method is knowledge distillation [13, 27, 17, 25, 12], which involves training a smaller network (student) to mimic the behaviour of a larger, more complex network (teacher) to achieve comparable performance with fewer computational resources. This idea has been investigated for compressing BNNs [18, 36, 30]. Although it is efficient to train the student in this manner, selecting the right teacher model which can guide the student during training may be difficult to a certain extent. In [18, 36, 30], the authors use teacher models, which are BNNs trained with Stochastic Gradient Langevin Dynamics (SGLD), resulting in prohibitive storage costs. Additionally, this knowledge

37th Conference on Neural Information Processing Systems (NeurIPS 2023).

distillation requires a multi-stage offline process involving pre-training the teacher, which could potentially limit the mutual learning between peer BNNs.

Mutual learning or online knowledge distillation [39, 40, 15, 4, 38] has received increasing interest in recent years. Mutual learning simplifies the training process of knowledge distillation to a single stage, which allows two or more networks to learn collaboratively and teach each other simultaneously, even with identical model architectures. In [39], the authors simply encourage multiple networks to learn from each other by reducing Kullback–Leibler (KL) divergence between pairs of predictions. In the same line of research, [40] proposes to train a multi-branch architecture by sharing low-level layers and concurrently creating an effective teacher based on soft logits to enhance the learning of the target network. Meanwhile, in [38, 4], the authors match the distribution of feature maps through adversarial training. In [4], the authors also indicate that a direct alignment method, such as the $L1$ or $L2$ distance between features of peer models could make networks become identical. Consequently, this does not improve the performance of peer networks in mutual learning. Even though these approaches show effective results, they mainly concentrate on the soft logits or features of peer networks, with limited research exploring the impact of diversity in the model parameter space for mutual learning. In the context of deep mutual learning, increasing diversity in parameter distributions and feature representations could encourage peer networks to learn distinct perspectives of data from one another. This process would enable models to acquire richer information, leading to more accurate decision-making.

In this work, we propose a novel approach to enhance the performance of BNNs by leveraging deep mutual learning. Specifically, we aim to increase diversity for deep mutual learning applied to BNNs. First, we propose increasing diversity in model parameter space by forcing posterior distributions to diversify. Next, we improve the BNNs' performance in the context of mutual learning by diversifying feature distributions. By promoting diversity in terms of model parameters and feature distributions we encourage peer networks to learn distinct features and characteristics from one another, ultimately boosting their overall performance. Our experimental results demonstrate that increasing diversity in both model parameter distributions and feature distributions is crucial for enhancing the performance of peer BNNs. We conduct experiments on the CIFAR-10, CIFAR-100, and ImageNet datasets and observe substantial improvements in classification accuracy, negative log-likelihood, and expected calibration error compared to the traditional mutual learning for BNNs. Our contributions can be summarized as follows:

- We propose a novel approach for improving the accuracy of BNNs by leveraging the effectiveness of mutual learning. Specifically, we propose promoting diversity in both model parameter space and feature space during the mutual training of peer BNNs. To our best knowledge, this is the first work that leverages mutual learning in the context of BNNs, and also investigates the usefulness of model parameter diversity in mutual learning. Additionally, we are the first to propose maximizing the distance between feature distributions to promote diversity in mutual learning for BNNs.

- We extensively evaluate our approach on CIFAR-10, CIFAR-100, and ImageNet datasets and demonstrate that it outperforms traditional mutual learning for BNNs in terms of accuracy, negative log-likelihood, and expected calibration error.

## 2   Related work

**Bayesian Neural Networks** have emerged as an active area of research due to their ability to capture uncertainty in predictions. While traditional deep learning models represent model parameters by point estimation and tend to be overconfident in their predictions [9], BNNs focus on estimating the posterior distribution over the weight parameters of neural networks. This approach can result in more calibrated predictions and better uncertainty estimations [1]. However, BNNs present several challenges, including intractable posterior distributions, unclear prior distributions, and a large number of parameters. To address these issues, researchers have proposed various methods for BNNs such as variational inference [1, 16, 5, 23], and Markov Chain Monte Carlo (MCMC) [22]. MCMC often results in a good performance, but this approach requires prohibitive computational cost and memory storage, making it impractical for real-world applications. In contrast, variational inference or mean-field variational inference (MFVI) presents a more convenient alternative by approximating the complicated posterior distribution with a simpler variational approximation. For instance, this

assumes a Gaussian posterior with diagonal covariance matrix [1]. Variational methods require storing only one model for evaluation, which is more scalable and useful in downstream tasks.

**Variational inference for Bayesian neural networks.** Given training dataset $\mathcal{S} = \{(x_n, y_n)\}_{n=1}^{N}$ with $x_n \in \mathcal{X}$ being an input and $y_n \in \mathcal{Y}$ being the corresponding output. The aim is to find a model $f(.; w) : \mathcal{X} \to \mathcal{Y}$, parameterized by $w \in \mathcal{W}$, that can accurately predict the output $y$ of each instance $x$. In Bayesian machine learning, that aim is equivalent to finding the posterior $p(w|\mathcal{S})$ of $w$ given the training dataset $\mathcal{S}$ with a prior $p(w)$ through the Bayes' theorem:

$$p(w|\mathcal{S}) = \prod_{n=1}^{N} \frac{p(y_n|x_n, w)\, p(w)}{\int p(y_n|x_n, w)\mathrm{d}w}. \tag{1}$$

The denominator in Eq. (1) is, however, intractable to be calculated exactly except for some simple models. To tackle this issue, variational inference is introduced to approximate the true posterior $p(w|\mathcal{S})$ by a variational posterior $q(w; \theta)$ parameterized by $\theta$. The variational posterior $q(w; \theta)$ can be obtained by minimizing the following KL divergence:

$$
\begin{aligned}
\theta^* &= \arg\min_{\theta} \mathrm{KL}[q(w; \theta) \| p(w|\mathcal{S})] \\
&= \arg\min_{\theta} \mathrm{KL}[q(w; \theta) \| p(w)] - \int q(w; \theta) \sum_{n=1}^{N} \log p(y_n|x_n, w)\mathrm{d}w \\
&= \arg\min_{\theta} \mathrm{KL}[q(w; \theta) \| p(w)] + \mathbb{E}_{q(w;\theta)}\left[ -\sum_{i=n}^{N} \log p(y_n|x_n, w) \right],
\end{aligned}
\tag{2}
$$

where $p(w)$ is the prior of the parameter $w$.

*Bayes by Backprop* (BBB) [1] is a simple but efficient variational approach. They suppose the approximate posterior following a Gaussian distribution with a diagonal covariance matrix, i.e., $q(w; \theta) = \mathcal{N}(\mu, \mathrm{diag}(\sigma^2))$ and apply a parametrization trick to represent each weight in the neural network by a mean and a standard deviation:

$$w = \mu + \sigma \odot \epsilon, \quad \epsilon \sim \mathcal{N}(0, I), \tag{3}$$

where: $\odot$ is the Hadamard product and $I$ is the identity matrix.

However, the standard BBB method in practice is sensitive to hyperparameters. To address this issue, in [5], the authors introduce a variant of BBB called Radial Bayesian Neural Network, where weight parameters are sampled as follows:

$$w = \mu + \sigma \odot \frac{\epsilon}{\|\epsilon\|} r, \quad r = |\rho| \quad \text{for} \quad \rho \sim \mathcal{N}(0, I), \tag{4}$$

where $\|.\|$ denotes the vector norm.

**Deep Mutual Learning (DML)**, also known as online knowledge distillation, has gained considerable interest recently due to its adaptability in various applications. Unlike offline knowledge which requires selecting and training a teacher network in advance, mutual learning involves distilling and sharing knowledge among peer networks. In [39], the authors argue that multiple models, even with identical architecture, can support each other during the learning process by minimizing the KL divergence between their soft logits. Regarding mutual learning in logits, ONE [40] generates an importance score for each student through the gate module, resulting in ensemble logits used as alignment targets for each network. In [10], optimal ensemble logits are computed and distilled among networks. However, these methods only utilize output logits without considering the potential benefits of incorporating knowledge from intermediate feature maps in peer networks. In [4], the authors show that directly minimizing the distance between features using $L1$ or $L2$ severely degrades the accuracy of peer networks, as it may cause models to learn in identical ways. Therefore for each network, in addition to learning useful features for the task on its own, [4] also encourages each network to learn the feature distribution of its peer network through an adversarial training strategy. By doing this, they encourage both networks to learn features that generalize better. In the other words, they encourage the two networks to learn features that work well for both, ultimately resulting in better performance. Despite these advancements, existing methods in mutual learning only use output logits or intermediate feature maps in peer networks and do not take advantage of knowledge from parameter space in peer networks, which could further promote diversity and play a crucial role in mutual learning.

**Bayesian Deep Mutual Learning.** Several works [36, 18, 30] adopt knowledge distillation methods to enhance the efficiency of BNNs. In [18], the authors train a teacher using SGLD [37] and then distil knowledge into a student network, which is a deterministic model. In [36], the authors attempt to reduce the storage issues of MCMC methods by distilling the posterior distribution of the BNN using the Generative adversarial networks (GANs). However, this approach could be difficult for larger models with millions of parameters. With this regard, in [30], the authors treat the knowledge in the teacher as prior to constraining the variational objective of the student. However, the aforementioned works still require getting MCMC samples in advance for distilling knowledge into variational BNNs, leading to expensive computational resources and large storage. This could be unsuitable for training large BNNs. In our work, we propose to apply mutual learning to improve the performance of variational Bayesian neural networks to make them applicable in real-work tasks. Specifically, to diversify features of peer BNNs, we propose to enlarge the distance between feature distributions of peer networks which is achieved by a careful design of feature combination, an appropriate choice of feature distribution approximation and a distance function. More importantly, in addition to diversity in feature space, we also investigate and leverage the optimal transport for encouraging diversity in model parameter space. To our best knowledge, this is the first work that investigates the usefulness of model parameter distribution diversity and also the first to enlarge the distance between feature distributions in mutual learning.

## 3 Proposed method

### 3.1 Deep Mutual Learning for Bayesian Neural Networks

We formulate our proposed method by considering two peer Bayesian neural networks, $B1$ and $B2$, with the goal of constructing an efficient framework for BNNs in mutual learning. The core idea of mutual learning [39] allows peer models to provide feedback on their learning to each other. This encourages peer models to acquire additional information from one another, ultimately leading to improved generalization performance. We adopt the two loss terms for logit-based learning; one is the conventional loss for variational Bayesian neural networks, and the other is mutual distillation loss between peer networks using KL divergence. In our setting, the approximate posterior distributions for each peer BNN is assumed to be a Gaussian distribution with a diagonal covariance matrix:

$$q(w; \theta_i) = \mathcal{N}(w; \mu_{\theta_i}, \text{diag}(\sigma_{\theta_i}^2)), \quad i = 1 : 2. \tag{5}$$

where $\mu_{\theta_i}, \sigma_{\theta_i} \in \mathbb{R}^d$, with $d$ denoting the number of model parameters.

The conventional loss for variational Bayesian neural network $B_i, i = 1 : 2$ is expressed as:

$$\mathcal{L}(\theta_i) = \text{KL}\left[q(w; \theta_i) \| p(w)\right] + \mathbb{E}_{q(w; \theta_i)}\left[-\sum_{n=1}^{N} \log p(y_n | x_n, w)\right]. \tag{6}$$

Let $z^{B1}$ and $z^{B2}$ be the logit outputs of the Bayesian neural networks $B1$ and $B2$, $\mathcal{T}$ be the temperature parameter to control the smoothness of the output distribution. We denote soft logit $p^{B1} = \text{softmax}(z^{B1}/\mathcal{T})$, and $p^{B2} = \text{softmax}(z^{B2}/\mathcal{T})$. The logit loss terms are defined as follows:

$$\mathcal{L}_{\text{logits}}^{B1} = \mathcal{L}(\theta_1) + \mathcal{T}^2 \times \text{KL}\left[p^{B2} \| p^{B1}\right], \tag{7}$$

$$\mathcal{L}_{\text{logits}}^{B2} = \mathcal{L}(\theta_2) + \mathcal{T}^2 \times \text{KL}\left[p^{B1} \| p^{B2}\right]. \tag{8}$$

### 3.2 Diversity in model parameter space

We aim to enhance diversity by leveraging the benefits of Bayesian neural networks over deterministic neural networks, in which weights in a BNN are represented by a distribution. Specifically, we increase the distance of posterior distributions and subsequently enhance the diversity in mutual learning. To the best of our knowledge, this is the first work that exploits the model parameter space to enhance diversity in mutual learning.

During the training process, for each iteration, we obtain Gaussian distributions with parameters $\theta_1 = (\mu_{\theta_1}, \Sigma_{\theta_1})$ and $\theta_2 = (\mu_{\theta_2}, \Sigma_{\theta_2})$ for BNN models $B1$ and $B2$, respectively. We then measure the distance between two posterior distributions $D(q(w; \theta_1), q(w; \theta_2))$. To this end, we exploit two well-known methods: Kullback-Liebler (KL) divergence and optimal transport. Both of them have

closed-form for Gaussian densities. While the KL divergence is well-known, it has limitations, such as being asymmetric and potentially infinite. Optimal transport theory [35] offers an alternative with the Wasserstein distance, which is symmetric and satisfies the triangle inequality. Furthermore, since the approximate posterior of the Bayesian neural network for the MFVI approach is Gaussian with diagonal covariance, this results in a simple form of optimal transport. Thus, we adopt the optimal transport, or Wasserstein distance, to measure the distance between two Gaussian distributions [34]:

$$D(q(w; \theta_1), q(w; \theta_2)) = \mathcal{W}_2(q(w; \theta_1), q(w; \theta_2))^2 = \|\mu_{\theta_1} - \mu_{\theta_2}\|^2 + \mathcal{B}(\Sigma_{\theta_1}, \Sigma_{\theta_2})^2, \quad (9)$$

where $\mathcal{B}(., .)$ is the Bures distance [2] represented by:

$$\mathcal{B}\left(\Sigma_{\theta_1}, \Sigma_{\theta_2}\right)^2 = \mathrm{tr}(\Sigma_{\theta_1} + \Sigma_{\theta_2} - 2(\Sigma_{\theta_1}^{1/2} \Sigma_{\theta_2} \Sigma_{\theta_1}^{1/2})^{1/2}). \quad (10)$$

With diagonal covariances $\Sigma_{\theta_1} = \mathrm{diag}(\sigma_{\theta_1}^2)$, and $\Sigma_{\theta_2} = \mathrm{diag}(\sigma_{\theta_2}^2)$, we have a simple form of distance between two approximate posteriors:

$$D(q(w; \theta_1), q(w; \theta_2)) = \|\mu_{\theta_1} - \mu_{\theta_2}\|^2 + \|\sigma_{\theta_1} - \sigma_{\theta_2}\|^2. \quad (11)$$

To promote diversity in the parameter space, a straightforward approach would be to add a negative $D(q(w; \theta_1), q(w; \theta_2))$ term to the objective functions in Eqs. (7) and (8) for minimization. However, doing so could lead to rapid increases of $D(q(w; \theta_1), q(w; \theta_2))$ which would impair the training process. Therefore, we propose minimizing the following loss function:

$$\mathcal{L}_{\text{diverse param}} = \log(1 + \exp(-D(q(w; \theta_1), q(w; \theta_2)))). \quad (12)$$

The above loss function exhibits desirable properties: $\mathcal{L}_{\text{diverse param}} \geq 0$, and minimizing $\mathcal{L}_{\text{diverse param}}$ implies maximizing $D(q(w; \theta_1), q(w; \theta_2))$, thus enhancing diversity in the model parameter space. These properties make optimization more stable during training.

### 3.3 Diversity in intermediate feature space

We aim to leverage intermediate features to increase the distance between the distributions of features among peer networks, rather than directly matching features using $L2$ distance as in offline distillation methods such as FitNets [27] and Attention transfer [17]. This is because directly matching features can lead to peer networks becoming identical, thereby reducing diversity in mutual learning. In our framework, the backbone networks are divided into the same blocks according to their depth. Let $F_k^{B1}$ and $F_k^{B2}$ be the extracted features at the $k^{th}$ block of the BNN models, $B1$ and $B2$, respectively. To increase the diversity of features among peer BNN models, we aim to maximize the distance between their feature distributions. It is worth noting that, diversifying peer networks' features is not a straightforward task. Our experimental results (detailed in the Supplementary Material) show that directly maximizing the $L_1$ or $L_2$ distance between peer networks' features does not necessarily enhance the effectiveness of mutual learning. Similarly, we observed the same outcome when directly maximizing the distance between peer networks' feature distributions. The reason could be that directly maximizing the distance between the distributions of original features $F_k^{B1}$ and $F_k^{B2}$ could potentially keep pushing the features apart indefinitely. This leads to BNN models becoming too dissimilar to effectively learn from each other, and then in turn negatively affect the predictive abilities of the BNN models. To mitigate this, we propose to diversify fused feature distributions. By doing so, we indirectly diversify feature distributions through transformed features, which allows for more flexible feature adaptation. Additionally, the use of fused features enables to increase diversity across multiple feature levels. We leverage cross-attention [33] to combine features from different blocks of each BNN. Specifically, given extracted features from the $k^{th}$ block, and $(k + 1)^{th}$ block of the network $B1$, the combined feature using cross-attention can be expressed as:

$$F'^{B1} = u(F_k^{B1}, F_{k+1}^{B1}) = \mathrm{softmax}\left[(\mathbf{W}_Q F_{k+1}^{B1})(\mathbf{W}_K F_k^{B1})^\top\right] \mathbf{W}_V F_k^{B1}, \quad (13)$$

where $\mathbf{W}_Q$, $\mathbf{W}_K$, and $\mathbf{W}_V$ represent learnable parameters for query, key, and value, respectively. Similarly, we have combined features $F'^{B2} = u(F_k^{B2}, F_{k+1}^{B2})$. Specifically, in a mini-batch with $n$ samples, we denote $G^{B1} = \{F_j'^{B1}\}_{j=1}^n$, and $G^{B2} = \{F_j'^{B2}\}_{j=1}^n$ represented fused features derived from BNN models $B1$ and $B2$, respectively.

Similar to the measurement of distance in the parameter space, we can utilize optimal transport. However the feature distributions may not be clearly represented by Gaussian distributions and

**Algorithm 1** Diversity for Bayesian Neural Networks in Mutual Learning.

---

1: **procedure** TRAIN($\mathcal{S}, n_{\text{batch size}}, n_{\text{epochs}}$)
2:    ▷ $\mathcal{S} = \{x, y\}$: *training dataset*                              ◁
3:    ▷ $n_{\text{batch size}}$: *size of a mini-batch*                            ◁
4:    ▷ $n_{\text{epochs}}$: *number of training epochs*                     ◁
5:    $\mu_{\theta_1}, \sigma_{\theta_1} \leftarrow$ RANDOM INITIALIZATION          ▷ *Initialize net B1*
6:    $\mu_{\theta_2} \leftarrow$ PRE-TRAINED MODEL                     ▷ *Initialize net B2*
7:    $\sigma_{\theta_2} \leftarrow$ RANDOM INITIALIZATION
8:    **for** epoch $= 1 \rightarrow n_{\text{epochs}}$ **do**
9:      **for** iter **in** iterations **do**
10:        sample a mini-batch: $\mathbb{B} = \{(x_i, y_i) : (x_i, y_i) \sim \mathcal{S}\}_{i=1}^{n_{\text{batch size}}}$

11:        ▷ *Compute losses shared between two peer networks*     ◁
12:        compute: $D_{\text{OT}} \leftarrow$ WASSERSTEIN DISTANCE($\mathcal{N}(\mu_{\theta_1}, \sigma_{\theta_1}^2), \mathcal{N}(\mu_{\theta_2}, \sigma_{\theta_2}^2)$) ▷ *Eq.* (11)
13:        compute diverse parameter distribution loss: $\mathcal{L}_{\text{diverse param}}$     ▷ *Eq.* (12)

14:        ▷ *Network B1*                                          ◁
15:        compute logit loss for B1: $\mathcal{L}_{\text{logits}}^{B1}$                      ▷ *Eq.* (7)
16:        compute diverse feature loss for B1: $\mathcal{L}_{\text{diverse feat}}^{B1}$         ▷ *Eq.* (16)
17:        compute total loss for B1: $\mathcal{L}^{B1} = \mathcal{L}_{\text{logits}}^{B1} + \alpha\mathcal{L}_{\text{diverse param}} + \beta\mathcal{L}_{\text{diverse feat}}^{B1}$  ▷ *Eq.* (18)
18:        update parameter: $\mu_{\theta_1}, \sigma_{\theta_1} \leftarrow$ ADAM($\mathcal{L}^{B1}$)

19:        ▷ *Network B2*                                          ◁
20:        compute logit loss for B2: $\mathcal{L}_{\text{logits}}^{B2}$                      ▷ *Eq.* (8)
21:        compute diverse feature loss for B2: $\mathcal{L}_{\text{diverse feat}}^{B2}$         ▷ *Eq.* (17)
22:        compute total loss for B2: $\mathcal{L}^{B2} = \mathcal{L}_{\text{logits}}^{B2} + \alpha\mathcal{L}_{\text{diverse param}} + \beta\mathcal{L}_{\text{diverse feat}}^{B2}$  ▷ *Eq.* (19)
23:        update parameter: $\mu_{\theta_2}, \sigma_{\theta_2} \leftarrow$ ADAM($\mathcal{L}^{B2}$)
24:    **return** $\mu_{\theta_1}, \sigma_{\theta_1}, \mu_{\theta_2}, \sigma_{\theta_2}$

---

other forms of optimal transport often require the computation of cost matrices, which are often computationally expensive. Therefore, we adopt KL divergence to diversify feature distributions. To this end, we generate the distribution of fused intermediate features by leveraging the conditional probability density of any two data points within the feature space [32], which is expressed as:

$$p_{i|j} = \frac{K(F_i', F_j')}{\sum_{k=1, k \neq i}^{n} K(F_k', F_j')}, \tag{14}$$

where $K(a, b)$ is a kernel function. In our work, we adopt the similarity metric $K(a, b) = \frac{1}{2}\left(\frac{a^T b}{\|a\|\|b\|} + 1\right)$ which has been used in [25] for offline knowledge distillation.

Given that $\mathcal{Q}$ and $\mathcal{P}$ are the probability distributions of fused features from BNN models $B1$ and $B2$, respectively, the distance between fused feature distributions is represented as:

$$D_{\text{KL}}(G^{B2}, G^{B1}) = \text{KL}\left[\mathcal{P}\|\mathcal{Q}\right] = \int \mathcal{P}(x) \log\left(\frac{\mathcal{P}(x)}{\mathcal{Q}(x)}\right) \mathrm{d}x. \tag{15}$$

It is worth noting that feature distribution has been investigated for offline knowledge distillation, i.e., in [25], the authors minimize KL divergence to transfer the probability distribution from teacher to student, rather than transferring actual representations. Conversely, in the context of mutual learning, our approach seeks to maximize the KL divergence between fused feature distributions to promote feature diversity. Similar to promoting diversity in the parameter space, we propose minimizing the following loss functions for BNN models $B1$, and $B2$:

$$\mathcal{L}_{\text{diverse feat}}^{B1} = \log\left(1 + \exp\left(-D_{\text{KL}}(G^{B2}, G^{B1})\right)\right), \tag{16}$$

$$\mathcal{L}_{\text{diverse feat}}^{B2} = \log\left(1 + \exp\left(-D_{\text{KL}}(G^{B1}, G^{B2})\right)\right). \tag{17}$$

Finally, the overall loss of our proposed Bayesian mutual learning for Bayesian neural networks $B1$ and $B2$ are defined as:

$$\mathcal{L}^{B1} = \mathcal{L}^{B1}_{\text{logits}} + \alpha\mathcal{L}_{\text{diverse param}} + \beta\mathcal{L}^{B1}_{\text{diverse feat}}, \quad (18)$$

$$\mathcal{L}^{B2} = \mathcal{L}^{B2}_{\text{logits}} + \alpha\mathcal{L}_{\text{diverse param}} + \beta\mathcal{L}^{B2}_{\text{diverse feat}}, \quad (19)$$

where $\alpha$, and $\beta$ are the hyper-parameters of total loss for optimization.

In our proposed BNNs in mutual learning strategy, the peer networks are jointly optimized and collaboratively learned during training. The optimization details are presented in Algorithm 1.

## 4 Experiments

### 4.1 Experimental setup

**Datasets.** We evaluate our approach on widely used datasets, including CIFAR-10 [20], CIFAR-100 [19], and ImageNet [29]. The CIFAR-100 dataset consists of 60,000 images across 100 classes, with 50,000 and 10,000 images used for training and validation, respectively. Meanwhile, the CIFAR-10 dataset has the same number of images for both training and validation sets but only consists of 10 classes. Additionally, we conduct experiments on ImageNet - a large and challenging dataset with 1,000 classes. This dataset contains about 1.2 million images for training and 50,000 images for validation, which we use as a test set in our experiments.

**Implementation details.** We evaluate the proposed BNNs-mutual learning with different ResNet models [11]. Specifically, on CIFAR-10 and CIFAR100 datasets, we evaluate our method with ResNet20, ResNet32 and ResNet56 architectures, while on the large scale ImageNet dataset, we evaluate our method with ResNet18 architecture. On CIFAR-10 and CIFAR-100, we train all models for 300 epochs, divided into two stages. In the first stage, we train peer networks in a mutual learning manner until convergence without the loss term that enhances diversity in features by setting the $\beta$ in Eq. 18 and Eq 19 to 0. Specifically, we train BNNs for up to 200 epochs. We use the Adam optimizer with an initial learning rate of 0.001, which is divided by 10 after 80, 120, 160, and 180 epochs. In the second stage, which comprises the last 100 epochs, we integrate the diversity in feature space into our BNNs-mutual learning and use the convergence from the first stage as an initial point for the training procedure. We continue training for the last 100 epochs in which the learning rate is reset to 0.001 and then is divided by 10 after 230, 260, and 290 epochs. The networks are divided into 3 blocks, and we combine features extracted from block 2 and block 3 of each BNN for feature diversity. In all experiments, the temperature $\mathcal{T}$ is set to 3 (Eq. 7 and Eq. 8), and $\alpha$ and $\beta$ (Eq. 18 and Eq. 19) are set to 1 and 2, respectively. On the ImageNet dataset, we train BNN models for up to 45 epochs. For the first 30 epochs, we set the $\beta$ to 0 and use the Adam optimizer with an initial learning rate of 0.001, which is divided by 10 after 15, 20, and 25 epochs. For the last 15 epochs, we reinitialize the learning rate to 0.001 and divide it by 10 after 35 and 40 epochs. We divide the networks into 4 blocks and combine features extracted from block 2, block 3, and block 4 of each BNN for feature diversity. The temperature $\mathcal{T}$, $\alpha$, and $\beta$ (Eq. 18 and Eq. 19) are set to 1. For all settings, we use a pair of Bayesian neural network models with the variational parameters to approximate posteriors (Eq. 6); one is initialized from scratch, and the other is initialized with the mean $\mu$ from the deterministic model trained on the same dataset. We empirically find that this setting achieves better results compared to peer models that are initialized under the same conditions. We sample a single network during the training phase, while for the inference phase, we conduct an ensemble of 50 sample models across all settings. All the reported results are average of 3 trials.

### 4.2 Experimental results

In this section, we compare our proposed method against deterministic models (DNN), Bayesian neural networks trained with mutual learning (DML) and trained without mutual learning (vanilla). In the vanilla approach, each BNN model is trained individually. We evaluate BNNs using three metrics including top-1 classification accuracy (ACC), negative log-likelihood (NLL), and expected calibration error (ECE) [9]. We first sample from the posterior distribution of the weights. We then average the predicted probabilities to calculate ACC and evaluate the negative log-likelihood of these probabilities for NLL. Meanwhile, to compute ECE, we categorize data samples into 20 bins based

Table 1: Top-1 classification accuracy, NLL, and ECE on CIFAR-100 dataset. *Bayesian neural networks are initialized with the mean value from the pre-trained deterministic model.

| | ACC ↑ | | | | NLL ↓ | | | | ECE ↓ | | | |
|---|---|---|---|---|---|---|---|---|---|---|---|---|
| | DNN | Vanilla | DML | Ours | DNN | Vanilla | DML | Ours | DNN | Vanilla | DML | Ours |
| ResNet20 | 69.13 | 63.18 | 67.27 | 68.32 | 1.106 | 1.447 | 1.174 | 1.101 | 0.065 | 0.119 | 0.057 | 0.041 |
| ResNet20* | - | 67.62 | 69.61 | **70.45** | - | 1.438 | 1.073 | **1.043** | - | 0.152 | 0.047 | **0.038** |
| Average | - | 65.40 | 68.44 | 69.39 | - | 1.443 | 1.124 | 1.072 | - | 0.136 | 0.052 | 0.040 |
| ResNet32 | 71.36 | 65.20 | 68.59 | 70.53 | 1.074 | 1.530 | 1.169 | 1.029 | 0.080 | 0.150 | 0.087 | 0.043 |
| ResNet32* | - | 69.76 | 71.45 | **72.14** | - | 1.542 | 1.012 | **0.975** | - | 0.173 | 0.064 | **0.040** |
| Average | - | 67.48 | 70.53 | 71.34 | - | 1.536 | 1.091 | 1.002 | - | 0.162 | 0.076 | 0.042 |
| ResNet56 | 72.88 | 66.60 | 70.44 | 71.90 | 1.146 | 1.853 | 1.117 | 1.024 | 0.153 | 0.204 | 0.092 | 0.066 |
| ResNet56* | - | 71.94 | 73.79 | **74.20** | - | 1.764 | 0.997 | **0.954** | - | 0.189 | 0.085 | **0.069** |
| Average | - | 69.27 | 72.12 | 73.05 | - | 1.809 | 1.057 | 0.989 | - | 0.197 | 0.089 | 0.068 |

Table 2: Top-1 classification accuracy, NLL, and ECE on CIFAR-10 dataset. *Bayesian neural networks are initialized with the mean value from the pre-trained deterministic model.

| | ACC ↑ | | | | NLL ↓ | | | | ECE ↓ | | | |
|---|---|---|---|---|---|---|---|---|---|---|---|---|
| | DNN | Vanilla | DML | Ours | DNN | Vanilla | DML | Ours | DNN | Vanilla | DML | Ours |
| ResNet20 | 91.83 | 89.93 | 91.09 | 92.03 | 0.302 | 0.445 | 0.295 | 0.256 | 0.067 | 0.066 | 0.040 | 0.029 |
| ResNet20* | - | 92.47 | 92.67 | **92.89** | - | 0.383 | 0.232 | **0.228** | - | 0.055 | 0.031 | **0.025** |
| Average | - | 91.20 | 91.88 | 92.46 | - | 0.414 | 0.264 | 0.242 | - | 0.061 | 0.036 | 0.027 |
| ResNet56 | 92.67 | 91.64 | 92.47 | 92.58 | 0.272 | 0.488 | 0.311 | 0.295 | 0.063 | 0.064 | 0.057 | 0.053 |
| ResNet56* | - | 93.33 | 93.50 | **93.91** | - | 0.431 | 0.243 | **0.237** | - | 0.054 | 0.041 | **0.039** |
| Average | - | 92.49 | 92.99 | 93.25 | - | 0.460 | 0.277 | 0.266 | - | 0.059 | 0.049 | 0.046 |

Table 3: Top-1 classification accuracy, NLL, and ECE on ImageNet dataset. *Bayesian neural networks are initialized with the mean value from the pre-trained deterministic model.

| | ACC ↑ | | | | NLL ↓ | | | | ECE ↓ | | | |
|---|---|---|---|---|---|---|---|---|---|---|---|---|
| | DNN | Vanilla | DML | Ours | DNN | Vanilla | DML | Ours | DNN | Vanilla | DML | Ours |
| ResNet18 | **69.76** | 64.30 | 66.88 | 67.36 | **1.247** | 1.505 | 1.357 | 1.343 | 0.130 | 0.134 | 0.134 | 0.131 |
| ResNet18* | - | 66.95 | 67.65 | 68.16 | - | 1.389 | 1.335 | 1.311 | - | 0.132 | 0.130 | **0.129** |
| Average | - | 66.63 | 67.27 | 67.76 | - | 1.447 | 1.346 | 1.327 | - | 0.133 | 0.132 | 0.130 |

on their confidence scores and measure the difference between the model's predicted probability and its actual accuracy.

**Comparative results on CIFAR-100 and CIFAR-10.** We present top-1 accuracy (%) (ACC), NLL, and ECE on both the CIFAR-100 and CIFAR-10 datasets for various peer BNNs, as shown in Tables 1 and 2. In terms of the average accuracy of peer BNNs in mutual learning, which represents the mean accuracy from the peer models, our proposed method consistently outperforms the BNNs trained with traditional mutual learning [39], with the highest improvement of 0.95% for ResNet20 on CIFAR-100 and 0.58% for ResNet20 on CIFAR-10, respectively. In terms of BNNs initialized with the mean value from the pre-trained deterministic model, our approach consistently outperforms the others on both datasets. The highest improvement is with ResNet20 architecture, i.e., our method outperforms DML 0.84% in terms of top-1 accuracy on the CIFAR-100 dataset. Regarding calibration results and negative log-likelihood, we also achieve better performance compared to other approaches across all pairs. Additionally, our BNN models that are initialized with the mean value from the pre-trained deterministic models outperform the deterministic models (DNN) in all metrics ACC, NLL, and ECE.

**Comparative results on ImageNet.** To further demonstrate the effectiveness of our proposed method, we conduct an experiment on the large-scale and complex ImageNet dataset, employing peer BNN models of ResNet18. As shown in Table 3, BNN networks trained with our method outperform those trained with DML [39], showing a gain of 0.49% for the average accuracy of two peer BNN networks and a gain of 0.51% for the BNN initialized with the mean value from the pre-trained deterministic model. This indicates that our method is applicable to large-scale datasets.

Table 4: Impact of diversity for each term in our proposed method in terms of top-1 classification accuracy. *Bayesian neural networks are initialized with the mean value from the pre-trained deterministic model.

| Setting | $\mathcal{L}_{\text{logits}}$ | $\mathcal{L}_{\text{diverse param}}$ | $\mathcal{L}_{\text{diverse feat}}$ | CIFAR100 | | |
| | | | | ResNet20 | ResNet20* | Average |
| --- | --- | --- | --- | --- | --- | --- |
| (a) | ✓ | | | 67.27 | 69.61 | 68.44 |
| (b) | ✓ | ✓ | | 67.78 | 70.22 | 69.00 |
| (c) | ✓ | | ✓ | 67.57 | 70.04 | 68.81 |
| (d) | ✓ | ✓ | ✓ | **68.32** | **70.45** | **69.39** |

Table 5: Comparison of top-1 classification accuracy on CIFAR-100 dataset when only partial data are retained based on the uncertainties of the predictions. *Bayesian neural networks are initialized with the mean value from the pre-trained deterministic model.

| Method | Model | 20% data retained | 40% data retained | 60% data retained | 80% data retained |
| --- | --- | --- | --- | --- | --- |
| Vanilla | ResNet32 | 98.90 | 92.75 | 83.22 | 74.34 |
| | ResNet32* | 99.10 | 95.38 | 87.43 | 78.29 |
| DML | ResNet32 | 99.20 | 94.83 | 86.33 | 77.43 |
| | ResNet32* | 98.95 | 95.43 | 87.73 | 80.10 |
| Ours | ResNet32 | 98.95 | 94.80 | 87.27 | 78.59 |
| | ResNet32* | **99.25** | **95.88** | **88.25** | **80.60** |

## 4.3 Ablation studies

**Impact of diversity in our proposed method.** In this section, we investigate how different diversity approaches in our proposed method contribute to the performance of variational BNNs in a mutual learning setting. All experiments are conducted on the CIFAR-100 dataset with peer BNNs using ResNet20. In Table 4, we present the top-1 accuracy of BNNs-mutual learning in various settings. The results show that when promoting diversity in parameter space, there are $0.51\%$ and $0.61\%$ improvements over the mutual learning that only uses $\mathcal{L}_{\text{logits}}$ for the BNNs initialized from scratch and initialized with the mean value from the deterministic neural network, respectively. When promoting diversity in feature space, the improvements are $0.3\%$ and $0.43\%$ over the same settings, respectively. This indicates that promoting diversity in parameter space has a more significant impact on model performance. By including both diversity in parameter and feature spaces, the highest improvement is achieved. This demonstrates that both loss terms are complementary to each other in achieving better performance in BNNs-mutual learning.

**The ability of uncertainty estimation.** To evaluate uncertainty estimation, we adopt Bayesian active learning by disagreement (BALD) [7], which quantifies the mutual information between the posterior distribution of parameters and the predictive distribution. We follow the evaluation methodology presented in Bayesian deep learning (BDL) benchmarks [6] to compare the accuracy in which only partial data are retained depending on uncertainty estimates obtained from the proposed method. Specifically, we compute uncertainty [7] for every testing image. Then we keep top images with the lowest uncertainty and report the top-1 accuracy of those kept images. This provides a more comprehensive evaluation metric of model performance and the ability to model uncertainty. Table 5 shows that accuracy increases when fewer data are retained on all networks. Furthermore, BNNs trained with our proposed method perform better than those trained without promoting diversity in mutual learning and those trained without mutual learning. This indicates that the proposed method produces more reliable uncertainty than the original mutual learning.

## 5 Conclusion

In this paper, we propose a novel approach to enhance BNNs by leveraging the effectiveness of mutual learning. Our approach encourages diversity in both model parameter space and feature space during the mutual learning of peer BNNs. By promoting such diversity, we encourage peer networks

to learn distinct features and perspectives from one another. This enhances the mutual learning of the peer networks. We extensively evaluate our method on CIFAR-10, CIFAR-100, and ImageNet datasets, demonstrating that the proposed method outperforms BNNs trained with and without mutual learning while producing reliable uncertainty estimates. We expect that our work will broaden the topic of mutual learning in which the model parameter space is taken into account, in addition to the traditional approaches that only consider the feature space.

## Acknowledgments

This work was partly supported by ARC DP23 grant DP230101176 and by the Air Force Office of Scientific Research under award number FA2386-23-1-4044.

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
