# Model and Feature Diversity for Bayesian Neural Networks in Mutual Learning Supplementary Material

**Cuong Pham**[1]      **Cuong C. Nguyen**[2]      **Trung Le**[1]      **Dinh Phung** [1,4]

**Gustavo Carneiro**[3]      **Thanh-Toan Do**[1]

[1]Department of Data Science and AI, Monash University, Australia

[2]Australian Institute for Machine Learning, University of Adelaide, Australia

[3]Centre for Vision, Speech and Signal Processing, University of Surrey, United Kingdom

[4]VinAI, Vietnam

`{cuong.pham1, trunglm, dinh.phung, toan.do}@monash.edu`

`cuong.nguyen@adelaide.edu.au, g.carneiro@surrey.ac.uk`

## A.1 Limitation

Even though our proposed approach improves the performance over traditional mutual learning, the proposed increasing diversity in parameter space is only applicable to identical Bayesian neural network models, limiting its generalizability to other Bayesian neural networks with different architectures.

## A.2 More ablation studies

### A.2.1 Method for diversity in feature space

Table A.1: Ablation studies on the CIFAR-100 dataset. Top-1 classification accuracy of mutual learning under various settings: (a) mutual learning applied without incorporating parameter and feature diversity; (b)-(c) direct maximization of distance between features using (b) L1, and (c) L2 norms; (d)-(e) Direct maximization of KL divergence between (d) distributions of features, and (e) distributions of fused features. *Bayesian neural networks are initialized with the mean value from the pre-trained deterministic model.

| Setting | ResNet20 | ResNet20* | Average | ResNet32 | ResNet32* | Average |
|:---:|:---:|:---:|:---:|:---:|:---:|:---:|
| (a) | 67.27 | 69.61 | 68.44 | 68.59 | 71.45 | 70.53 |
| (b) | 67.37 | 69.58 | 68.50 | 70.21 | 71.65 | 70.93 |
| (c) | 67.32 | 69.50 | 68.41 | 70.42 | 71.76 | 71.09 |
| (d) | 67.74 | 69.65 | 68.70 | 70.32 | 71.89 | 71.11 |
| (e) | 67.57 | **70.04** | 68.81 | 70.35 | **71.95** | 71.15 |

We evaluate top-1 classification accuracy under various mutual learning settings, by conducting a series of ablation studies on the CIFAR-100 dataset, as shown in Table A.1. Those settings comprise several aspects. The first is traditional mutual learning without incorporating parameter and feature diversity. Secondly, we explore the direct maximization of the distance between features using $L1$ or $L2$ norms. We also test the direct maximization of Kullback-Leibler (KL) divergence between feature distributions. Finally, we experiment with the proposed approach that made use of fused

37th Conference on Neural Information Processing Systems (NeurIPS 2023).

feature distributions. The goal of these ablation studies is to gain a more understanding of how these individual methods influence the performance of mutual learning.

**Direct maximize distance between features with $L1$, $L2$ distance.** As presented in Table A.1, the direct maximization of $L1$ or $L2$ distance between the features of peer Bayesian neural networks does not necessarily improve the effectiveness of mutual learning. Specifically, traditional mutual learning as presented in setting (a) outperforms the methods that explicitly maximize the $L1$ (as in setting b) or $L2$ distance (as in setting c) between peer networks' features. This indicates that maximizing these distances may not be the best strategy for enhancing mutual learning.

**Direct maximize KL divergence between feature distributions.** We further conduct ablation studies focusing on directly maximizing the Kullback-Leibler (KL) divergence between feature distributions of peer Bayesian neural networks (as in setting d in Table A.1). This approach offers a slight improvement over strategies that utilized $L1$ or $L2$ distance. However, the performance achieved is lower than when maximizing the distance between fused feature distributions (as in setting e in Table A.1).

### A.2.2 Method for diversity in parameter space

Table A.2: Ablation studies on the CIFAR-100 dataset. Comparison of top-1 classification accuracy of mutual learning when measuring distance between parameter distributions using Kullback-Leibler (KL), and Optimal transport (OT). *Bayesian neural networks are initialized with the mean value from the pre-trained deterministic model.

| Setting | ResNet20 | ResNet20* | Average | ResNet32 | ResNet32* | Average |
|---------|----------|-----------|---------|----------|-----------|---------|
| KL | 67.44 | 69.98 | 68.71 | 69.65 | 71.80 | 70.73 |
| OT | 67.78 | **70.22** | 69.00 | 69.70 | **71.99** | 70.85 |

**KL vs OT in parameter space.** We conduct experiments to compare the use of optimal transport and Kullback-Leibler (KL) divergence for measuring distances in the parameter space. As shown in Table A.2, the results for both ResNet20 and ResNet32 BNN models demonstrate that using optimal transport yields higher accuracy compared to using KL divergence.

## A.3 The choice of hyper parameters

Regarding hyper parameters $T$, $\alpha$, $\beta$: For the temperature $T$, we follow the seminal works [2] and [1]. This parameter $T$ controls the smoothness of the prediction distribution. As the value of $T$ increases, the prediction distribution becomes smoother. The hyper parameters $\alpha$ and $\beta$ control the impacts of the diversity of network parameter distributions and network feature distributions on the learning of a pair of peer networks. We present the ablation studies on the choice of $T$, $\alpha$, and $\beta$ on CIFAR100. "*" means Bayesian neural networks that are initialized with the mean value from the pre-trained deterministic model.

For ablation studies for parameter T, we vary the value of $T$ from 1 to 5 and fix the values of $\alpha = 1$ and $\beta = 2$. The results are shown in Table A.3. The results show that the best value of $T$ is 3.

Table A.3: Ablation studies for parameter T.

| T | 1 | 2 | 3 | 4 | 5 |
|---|---|---|---|---|---|
| ResNet20 | 66.6 | 67.66 | 68.32 | 67.95 | 67.93 |
| ResNet20* | 69.62 | 70.14 | 70.45 | 70.12 | 69.97 |

For ablation studies for parameter $\alpha$, we vary $\alpha$ when promoting diversity in parameter space, and set the value of $T = 3$ and $\beta = 0$. The results are shown in Table A.4. The results show that by setting $\alpha = 1$, we achieve a better performance compared to other tested values of $\alpha$.

Table A.4: Ablation studies for parameter $\alpha$.

| $\alpha$ | 0.1 | 1 | 2 | 5 | 10 |
|---|---|---|---|---|---|
| ResNet20 | 67.25 | 67.78 | 67.18 | 67.45 | 67.29 |
| ResNet20* | 69.79 | 70.22 | 69.92 | 69.61 | 69.48 |

For ablation studies for parameter $\beta$, we vary $\beta$ when promoting diversity in feature space, and set the value of $T = 3$ and $\alpha = 0$. The results are shown in Table A.5. The results show that by setting $\beta = 2$, we achieve a better performance compared to other tested values of $\beta$.

Table A.5: Ablation studies for parameter $\beta$.

| $\beta$ | 0.1 | 1 | 2 | 5 | 10 |
|---|---|---|---|---|---|
| ResNet20 | 66.95 | 67.17 | 67.57 | 67.14 | 67.09 |
| ResNet20* | 69.84 | 69.91 | 70.04 | 69.85 | 69.90 |

In addition, from our experiments, we found that the accuracies are only slightly different when $\alpha$ and $\beta$ take values in $\{1, 2\}$.

## A.4 Visualizations

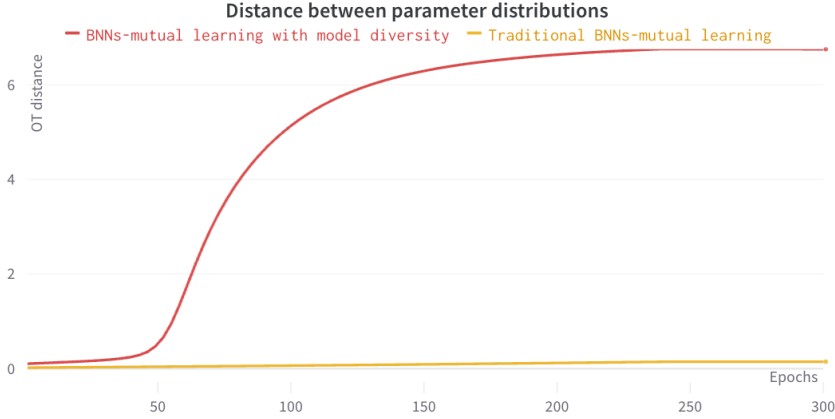

Figure A.1: Comparison of optimal transport distance between the parameter distributions of peer Bayesian neural networks in mutual learning with diversity in parameter space and traditional mutual learning [3] on the CIFAR-100 dataset.

**Distance between parameter distributions.** For each iteration, we obtain Gaussian distributions with parameters $\theta_1 = (\mu_{\theta_1}, \Sigma_{\theta_1})$ and $\theta_2 = (\mu_{\theta_2}, \Sigma_{\theta_2})$ for BNN models $B1$ and $B2$, respectively. We then measure the distance between two posterior distributions $D(q(w; \theta_1), q(w; \theta_2))$. With diagonal covariances $\Sigma_{\theta_1} = \mathrm{diag}(\sigma^2_{\theta_1})$, and $\Sigma_{\theta_2} = \mathrm{diag}(\sigma^2_{\theta_2})$, we have a simple form of distance between two approximate posteriors:

$$D(q(w; \theta_1), q(w; \theta_2)) = \|\mu_{\theta_1} - \mu_{\theta_2}\|^2 + \|\sigma_{\theta_1} - \sigma_{\theta_2}\|^2. \tag{1}$$

During the training process, we measure the optimal transport distance between parameter distributions using Eq. 1. As shown in Figure. A.1, it is clear that our proposed method, which promotes diversity in the parameter space, leads to larger distances between parameter distributions compared to traditional mutual learning. Additionally, we observe that the optimal transport distance between parameter distributions in our proposed method increases only until it reaches a saturation point. This indicates that while our method maximizes the distance between parameter distributions, this distance does not grow indefinitely.

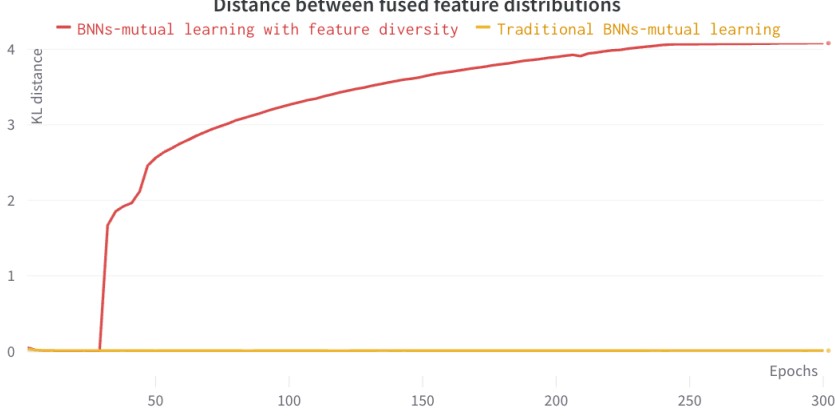

Figure A.2: Comparison of KL divergence between the fused feature distributions of peer Bayesian neural networks in mutual learning with diversity in feature space and traditional mutual learning on the CIFAR-100 dataset.

**Distance between fused feature distributions.** Given that $\mathcal{Q}$ and $\mathcal{P}$ are the probability distributions of fused features from BNN models $B1$ and $B2$ (detailed in the main paper), respectively, the distance between fused feature distributions is represented as:

$$D_{\mathrm{KL}}(G^{B2}, G^{B1}) = \mathrm{KL}\left[\mathcal{P}\|\mathcal{Q}\right] = \int \mathcal{P}(x) \log\left(\frac{\mathcal{P}(x)}{\mathcal{Q}(x)}\right) \mathrm{d}x. \tag{2}$$

During the training process, we measure the KL divergence between fused feature distributions. As shown in Figure. A.2, it is clear that our proposed method, which promotes diversity in the feature space, leads to larger KL divergence between fused feature distributions compared to traditional mutual learning. Additionally, we note that in our proposed method, the KL divergence between fused feature distributions increases only until it reaches a saturation point. This indicates that while our method maximizes the distance between fused feature distributions, this distance does not grow indefinitely.