# OpenReview forum: "Model and Feature Diversity for Bayesian Neural Networks in Mutual Learning"
_NeurIPS.cc/2023/Conference — NeurIPS 2023 poster_

### Official Review · Reviewer_5v8y · 2023-07-01

**Soundness:** 3 good
**Presentation:** 3 good
**Contribution:** 2 fair
**Rating:** 5
**Confidence:** 2

**Summary:**

This paper proposed a mutual learning approach to learn a pair of Bayesian Neural Network(BNN). The posterior of BNN is approximated by Variational Inference using a Gaussian distribution with a diagonal covariance matrix. To make the BNN learn different perspective of the data, the author proposed to increase the diversity in parameter space and intermediate feature space by adding the an estimate of distance between parameter distribution and fused feature distribution of two BNN models into the objective function. Empirically, the proposed method outperform existing mutual learning method and vanilla BNN model in terms of accuracy, negative log likelihood loss and expected calibration error. An ablation study is also provided to investigate the usefulness of each component.

**Strengths:**

The paper is well written and easy to follow. Increasing the diversity of parameter distribution and intermediate feature distribution of peer BNN models to boost performance is an interesting idea. Experiments and detailed ablation study demonstrate the effectiveness of proposed method.

**Weaknesses:**

1. It is mentioned in the abstract and introduction that the BNN model with variational inference may underperform deterministic model or BNN obtained by MCMC, the baseline only involves BNN model trained with(DML) or without(vanilla) mutual learning. Would the proposed method close the gap to some extent? Data augmentation, optimizer may all affect performance, so it is still helpful to include deterministic model results follow with same training setup. I would expect the BNN model to outperform deterministic model at least in NLL and ECE, and with the 50 ensemble, it can outperform the accuracy.

2. Continue with last point, for MCMC method (e.g. in line 81 of the paper), I agree that traditional MCMC method(e.g. Metropolis Hasting) may not be feasible for large model, and memory storage can be an issue for MCMC method. But I don't think the stochastic gradient MCMC cited in line 81 would require prohibitive computational cost, it behaves like adding a noise to at each step of standard SGD training.

3. The code is not provided so it may hurt the reproducibility of the paper.

**Questions:**

1. To my knowledge, it is not very clear if variational distribution(e.g. Gaussian with diagonal covariance matrix) can approximate the true posterior very well, can the author comment a bit on this, e.g. how would different choice of variational family affect the model?

2. In line 264 and line 6 of algorithm 1, it is mentioned that one BNN model is initialized with a trained model and this lead to better results empirically. Can the author discuss more on why this happened? It is a bit wired for me as it seems in the implementation detail, the pre-trained model and the model from scratch are trained with same optimizer and learning rate schedule.

3. Seems like $\alpha$ $\beta$ are set to 1,2 for CIFAR and 1,1 for Imagenet, these two parameters controls the strength of proposed penalty to the model, can the author comments a bit more on how sensitive are the model to those parameters, It can help to illustrate how diversity helps model performance.

4. As mentioned in line 268, results are average of 3 trials, I think it would be better to include the standard deviation as well to boost the significance of the results.

5. In figure A.3 in supplementary material, looks like a sharp increase of KL divergence between the fused feature distributions at around 30 epochs, but the penalty for feature is only added for last 100 epochs, can the author explain more on this?

**Limitations:**

The authors addressed the limitations.

---

> ### Author Rebuttal · Authors · 2023-08-09
>
> ### Weaknesses
>
> 1. BNN model with variational inference may underperform deterministic model or BNN obtained by MCMC, the baseline only involves BNN model trained with (DML) or without(vanilla) mutual learning. Would the proposed method close the gap to some extent? I would expect the BNN model to outperform deterministic model at least in NLL and ECE, and with the 50 ensemble, it can outperform the accuracy.
>
> **Answer:**  Due to limited space we cannot add tables here. We compare our method with deterministic models and also with MCMC-based BNNs. For MCMC-based BNNs, we use the Stochastic Gradient Langevin Dynamics (SGLD). The results are presented in Table 1 in the attached pdf file which can be found at the end of our general response at the beginning.  The experimental results show that our proposed method outperforms the deterministic model and the SGLD [36].
>
> 2. I don't think the stochastic gradient MCMC cited in line 81 would require prohibitive computational cost, it behaves like adding a noise to at each step of standard SGD training.
>
> **Answer:**  We agree with Reviewer that SGLD [3] and Stochastic Gradient Hamilton Monte Carlo (SGHMC) would not require prohibitive computational cost. We will revise the references. We also would like to note that comparing to the stochastic gradient based MCMC such as SGLD and SGHMC, Variational Inference (VI)-based BNNs is more efficient in term of memory storage. In MCMC-based BNNs, we need to store model samples, while in VI-based BNNs, we can explicitly estimate the posterior.
>
> 3. The code is not provided so it may hurt the reproducibility of the paper?
>
> **Answer:**  We will release the code for the reproducibility of the paper.
>
> ### Questions
>
> 1. To my knowledge, it is not very clear if variational distribution (e.g. Gaussian with diagonal covariance matrix) can approximate the true posterior very well, can the author comment a bit on this, e.g. how would different choice of variational family affect the model?
>
> **Answer:** The accuracy of the approximation made in VI mostly depends on how expressive the variational family is. The more expressive, the better the approximation. However, in practice, one needs to balance between the accuracy of the approximation and the computation and storage complexity associated. One of the simplest variational families includes the Gaussian distribution with diagonal covariance matrix [1] or Bernoulli distribution [A]. Despite their modest in terms of posterior estimation, they are widely used in the literature due to their efficiency (e.g., closed-form KL divergence) and scalability (i.e., the number of parameters scales linearly) which is suitable for large-scale models such as deep neural networks. Subsequently, one can select a family with higher complexity and expressiveness, such as Gaussian mixture models, to approximate the true posterior accurately. Although such a model could approximate the true posterior precisely, it significantly increases the number of learnable parameters, requiring more computation and storage, and hence, making it less applicable for deep neural networks.
>
> [A] Yarin Gal and Zoubin Ghahramani, Bayesian Convolutional Neural Networks with Bernoulli Approximate Variational Inference, ICLR Workshops, 2016.
>
> 2. In line 264 and line 6 of algorithm 1, it is mentioned that one BNN model is initialized with a trained model and this lead to better results empirically. Can the author discuss more on why this happened? It is a bit wired for me as it seems in the implementation detail, the pre-trained model and the model from scratch are trained with same optimizer and learning rate schedule.
>
> **Answer:** One of the benefits of using the pretrained model as initialization for the mean of BNNs is that the pretrained model has been well trained. Using it for BNNs training will lead to better training convergence. In addition, as our aim is to make the model distributions and feature distributions of two peer networks diverse, having a BNN initialized with a pretrained model and a BNN initialized from scratch at the beginning of the training ensures the diversity in model parameter distributions and feature distributions. The above benefits lead to better results.
>
> 3. Seems like $\alpha$, $\beta$ are set to 1,2 for CIFAR and 1,1 for Imagenet, these two parameters controls the strength of proposed penalty to the model, can the author comments a bit more on how sensitive are the model to those parameters, It can help to illustrate how diversity helps model performance.
>
> **Answer:** Please refer to the general response.
>
> 4. As mentioned in line 268, results are average of 3 trials, I think it would be better to include the standard deviation as well to boost the significance of the results.
>
> **Answer:** We present in Table 2 in the attached pdf file the ACC, NLL, and ECE with the standard deviation of pairs of networks on CIFAR100 dataset.
>
> 5. In figure A.3 in supplementary material, looks like a sharp increase of KL divergence between the fused feature distributions at around 30 epochs, but the penalty for feature is only added for last 100 epochs, can the author explain more on this?
>
> **Answer:** We would like to clarify that for figure A.3 in the supplementary material, we train BNNs with only $L_{\text{logits}}$ and $L_{\text{diverse feat}}$, i.e., we do not add the model diversity. This corresponds to the setting 'c' in Table 4 in the main paper. Our aim is to show that when diversity is encouraged in the feature space, the resulting distance between fused feature distributions is greater than the distance observed during when training with the traditional mutual learning.  We also would like to note that figure A.2 is with training BNNs with only $L_{\text{logits}}$ and $L_{\text{diverse param}}$ which corresponds to the setting 'b' in Table 4 in the main paper.

---

> > ### Comment · Reviewer_5v8y · 2023-08-14
> >
> > I appreciate the authors for detailed response. I will keep my score.

---

> > > ### Author Response · Authors · 2023-08-15
> > >
> > > Dear Reviewer 5v8y,
> > >
> > > Thank the Reviewer for the feedback. We greatly appreciate the time and effort the Reviewer dedicated to considering our paper and our response.

---

### Official Review · Reviewer_fyNb · 2023-07-02

**Soundness:** 3 good
**Presentation:** 3 good
**Contribution:** 2 fair
**Rating:** 6
**Confidence:** 3

**Summary:**

The paper proposes a method to combine deep mutual learning with BNN to diversify the weight distributions of each BNN networks in a pair or ensemble, to improve performance.

**Strengths:**

1. AFAIK this is the first work combining mutual learning with BNN, so the authors can claim this point.
2. The paper is in general written clearly and easy to follow.
3. Experiments are adequate with ablation studies on individual features impact on diversity.

**Weaknesses:**

1. Some design choices are found to be "empirically" working well without too much discussion or hypothesis.
2. Would be interesting to see how the model performs for o.o.d test data, especially uncertainty performance.

**Questions:**

Line 178-179: The authors said adding the D(...) term will rapidly increase of this term and impact training. Wouldn't putting a smaller scaling factor for this term fix this issue?

**Limitations:**

None.

---

> ### Author Rebuttal · Authors · 2023-08-09
>
> ### Weaknesses
>
> 1. Some design choices are found to be "empirically" working well without too much discussion or hypothesis.
>
> **Answer:** Please refer to the general response for the ablation studies on hyper parameters T, $\alpha$, and $\beta$.
>
> 2. Would be interesting to see how the model performs for o.o.d test data, especially uncertainty performance.
>
> **Answer:**  We thank the Reviewer for the suggestion. To address this, we trained BNN models on the first 5 classes of the CIFAR10 dataset, referred to as CIFAR5. The data from the remaining 5 classes are considered as OOD data. To estimate the uncertainty of predictions on OOD datasets, we compute the entropy of the average prediction from 50 sampled models (for each BNN), called OOD Entropy.
>     We present the top-1 classification accuracy, evaluated on the first 5 classes, as well as the uncertainty estimation on the corresponding OOD data. The comparisons between our proposed method, the DNN model, the vanilla BNN, and the BNN with DML are presented in the following table. The results show that, when trained on CIFAR5, our method outperforms the others in terms of accuracy and uncertainty on OOD data. DNN yields the lowest uncertainty on OOD data, while our proposed method yields the highest uncertainty on OOD data.
>
> Table: Top-1 classification accuracy on CIFAR5 dataset and entropy on corresponding OOD data. *Bayesian neural networks are initialized with the mean value from the pre-trained deterministic model. We report the mean and standard deviation over 3 runs.
>
> | Method  | Model       | ACC (%)                    | OOD Entropy                    |
> |---------|-------------|----------------------------|--------------------------------|
> | DNN     | ResNet20    | 89.46 ± 0.08               | 0.209 ± 0.028                  |
> | Vanilla BNN | ResNet20    | 84.27 ± 0.42               | 0.244 ± 0.030                  |
> |         | ResNet20*   | 90.35 ± 0.45               | 0.261 ± 0.015                  |
> | DML     | ResNet20    | 86.74 ± 0.32               | 0.296 ± 0.005                  |
> |         | ResNet20*   | 91.06 ± 0.32               | 0.325 ± 0.010                  |
> | Ours    | ResNet20    | 86.89 ± 0.12               | 0.308 ± 0.019                  |
> |         | ResNet20*   | **91.36 ± 0.25**           | **0.358 ± 0.022**
>
> ### Questions
>
> 1. Line 178-179: The authors said adding the D(...) term will rapidly increase this term and impact training. Wouldn't putting a smaller scaling factor for this term fix this issue?
>
>  **Answer:** In our preliminary experiments, we tried with adding a scale factor to control $D(q(w; \theta_1), q(w; \theta_2))$, however, we found that it is not an effective solution. The reason is that D(.,.) does not have an upper bound, which implies that it is not easy to control the scaling factor. The value of the D(.,.) term could potentially increase indefinitely during the learning process if we directly maximize it. This could lead to gradient explosion and the model will not converge. The proposed loss $L_{diverse param}$ (eq. 12) has the advantage that the D(.,.) increases gradually until it reaches a saturation point. These properties make optimization more stable during training.

---

> > ### Comment · Reviewer_fyNb · 2023-08-18
> >
> > Thanks for the additional results. I will keep my score.

---

> > > ### Author Response · Authors · 2023-08-18
> > >
> > > Dear Reviewer fyNb,
> > >
> > > We thank the Reviewer for the feedback. We greatly appreciate the time and effort the Reviewer dedicated to considering our paper and our response.

---

> > ### Comment · Reviewer_fyNb · 2023-08-22
> >
> > Thanks authors for the reply. I decide to keep my score of weak accept.

---

### Official Review · Reviewer_phoh · 2023-07-05

**Soundness:** 2 fair
**Presentation:** 3 good
**Contribution:** 2 fair
**Rating:** 5
**Confidence:** 3

**Summary:**

The paper titled addresses the challenge of improving the performance of Bayesian Neural Networks (BNNs) by leveraging the concept of mutual learning. BNNs provide a means for quantifying uncertainty in predictions through probability distributions of model parameters. However, BNNs often fall short in performance compared to their deterministic counterparts. The authors propose a novel approach that employs deep mutual learning to enhance the capabilities of BNNs.

**Strengths:**

1. Innovative Approach: The paper introduces a novel method that combines deep mutual learning with Bayesian Neural Networks. By promoting diversity in both network parameter distributions and feature distributions, the proposed approach enables peer networks to acquire distinct features, capturing different characteristics of the input data. This innovative technique enhances the effectiveness of mutual learning in BNNs.
2. Detailed algorithm description: The paper provides a thorough and detailed description of the proposed algorithm for improving the performance of Bayesian Neural Networks (BNNs) through deep mutual learning.
3. Comprehensive Experiments: The authors conduct extensive experiments to evaluate the proposed approach thoroughly. The experimental results are statistically sound and demonstrate significant improvements in classification accuracy, negative log-likelihood, and expected calibration error compared to traditional mutual learning methods for BNNs.

**Weaknesses:**

1. Limited variety in experimental validation: One weakness of the paper is that the proposed approach and its effectiveness are only verified through experiments conducted on Residual Neural Networks (ResNets). It would have been beneficial to include experiments on a diverse set of network architectures to demonstrate the approach's effectiveness across different model types and complexities.
2. Lack of detailed explanation for temperature, α, and β: One weakness of the paper is the limited explanation provided for the temperature parameter (T), α, and β, which are crucial components of the proposed approach. These parameters play a significant role in controlling the diversity of network parameter distributions and feature distributions, but their specific effects and optimal values are not thoroughly discussed.
3. Weakness in the conclusion: The current conclusion merely restates the experimental results and does not highlight the broader implications of the proposed approach or its potential impact on the field.

**Questions:**

The author should supplement more experiments to prove its effectiveness.

**Limitations:**

The author should supplement more experiments to prove its effectiveness and strengthen the conclusion.

---

> ### Author Rebuttal · Authors · 2023-08-09
>
> ### Weaknesses
> 1. Limited variety in experimental validation: One weakness of the paper is that the proposed approach and its effectiveness are only verified through experiments conducted on Residual Neural Networks (ResNets). It would have been beneficial to include experiments on a diverse set of network architectures to demonstrate the approach's effectiveness across different model types and complexities.
>
> **Answer:** In addition to evaluations with the ResNet architectures in the main paper, here we validate our proposed method on AlexNet. The following table shows the results of ACC, NLL, and ECE. In terms of BNNs initialized with the mean value from the pre-trained deterministic model, our approach consistently outperforms the others. Our method outperforms DML 0.48\% and DNN 0.92\% in terms of  top-1 accuracy on the CIFAR-100 dataset.
>
> Table: Top-1 classification accuracy, NLL, and ECE on CIAFR100 dataset. DNN means the deterministic model. *Bayesian neural networks are initialized with the mean value from the pre-trained deterministic model.
>
> |          |               | ACC↑     |           |           |               | NLL↓     |           |           |               | ECE↓     |           |           |
> |----------|---------------|-----------|-----------|-----------|---------------|-----------|-----------|-----------|---------------|-----------|-----------|-----------|
> |          | Vanilla       | DNN       | DML       | Ours      | Vanilla      | DNN       | DML       | Ours      | Vanilla      | DNN       | DML       | Ours      |
> | AlexNet  | 50.47         | 52.40     | 51.82     | 52.10     | 2.325         | 2.141     | 2.255     | 2.230     | 0.187         | 0.142     | 0.172     | 0.165     |
> | AlexNet* | 52.23         | -         | 52.84     | 53.32     | 2.105         | -         | 1.921     | 1.904     | 0.169         | -         | 0.099     | 0.089     |
>
> 2. Lack of detailed explanation for temperature, $\alpha$, and $\beta$: One weakness of the paper is the limited explanation provided for the temperature parameter (T),$\alpha$, and $\beta$, which are crucial components of the proposed approach. These parameters play a significant role in controlling the diversity of network parameter distributions and feature distributions, but their specific effects and optimal values are not thoroughly discussed.
>
> **Answer:** Please refer to the general response.
>
> 3. Weakness in the conclusion: The current conclusion merely restates the experimental results and does not highlight the broader implications of the proposed approach or its potential impact on the field.
>
> **Answer:**  Our work is the first work that explores the potential of deep mutual learning in the context of BNNs. More importantly, we are also the first to investigate the usefulness of model parameter diversity in mutual learning. We expect that our work will broaden the topic of mutual learning and inspire further researches in which the model parameter space is taken into account, in addition to the traditional approaches that only consider the feature space. In addition, our work can be considered as a baseline for DML-BNNs and we expect that our work inspires further researches that investigate the usefulness of mutual learning in the classical field BNNs. We will update our conclusion to reflect the above.
>
>
> ### Questions
> 1. The author should supplement more experiments to prove its effectiveness.
>
> **Answer:**  We present above the experiments with AlexNet architecture. We also compare our work with the work [4] when [4] is applied to BNNs. We also compare our approach with deterministic models (i.e., single deterministic models and size-2 deep ensemble models). The results are presented in the tables in our responses to Reviewer xPiJ and Reviewer tB5V.

---

> > ### Comment · Reviewer_phoh · 2023-08-20
> >
> > I'm grateful for your response that tackled my concern. It's satisfying to witness its efficacy confirmed on more widely used deep network architectures. Consequently, I've opted to retain my score.

---

> > > ### Author Response · Authors · 2023-08-20
> > >
> > > Dear Reviewer phoh,
> > >
> > > We thank the Reviewer for the feedback. We greatly appreciate the time and effort the Reviewer dedicated to considering our paper and our response.

---

### Official Review · Reviewer_tB5V · 2023-07-06

**Soundness:** 2 fair
**Presentation:** 3 good
**Contribution:** 2 fair
**Rating:** 4
**Confidence:** 4

**Summary:**

The paper focuses on improving the accuracy of BNNs by promoting diversity in both parameter space and feature space while training two peer BNNs with mutual learning between them. More specifically, they train two variational BNNs with a mean-field Gaussian variational loss for each along with a KL divergence term between the (temperature-scaled) predictive distributions of the two models, a Wasserstein distance term between the corresponding approximate posterior distributions across the two models (added as a softplus(-distance) term), and a KL divergence term between corresponding feature distributions. On the latter term, instead of directly maximizing the distance between corresponding feature distributions, they instead do so on "fused feature distributions". To do so, they use learned cross-attention to fuse the features from multiple feature levels in a model (two at a time). Then, they use the KL divergence between the distributions of the fused feature distributions of the two peer networks. To derive the distributions, they use the conditional probability density defined as $p_{i|j} = \frac{K(F'_i, F'_j)}{sum_{k=1, k \noteq i}^n K(F'_k, F'_j)}$, where $K(F'_a, F'_b)$ is a kernel function between two fused feature representations. Given those conditional probs, they compute a KL divergence term. Similar to the parameter space diversity term, they add this term to the loss as softplus(-divergence). The paper claims to be the first to propose maximizing the distance between feature distributions to promote diversity. In terms of experiments, the paper includes results for ResNet models on CIFAR-10/100 and ImageNet, measuring accuracy, NLL, and ECE as metrics, and comparing different approaches.

**Strengths:**

The paper does a great job of precisely articulating the modeling approach, and discussing the relevant background info. More specifically, the proposed approach of adding terms to promote diversity in parameter and feature space is clear and would be easy to reimplement.

**Weaknesses:**

My main concern is with the experiment section. More specifically, a few key details are unclear in the text, and importantly a deterministic baseline is missing that I believe should be present given the framing of the paper and relevant literature. Please see the Questions below. Given updates, I believe the paper would be great and I would gladly update my rating.

**Questions:**

Main:
- In the experiments, a few details are currently unclear. The following points are on Table 1, but generalize to all three tables. Please clarify these details here and in the paper.
  - Consider the ResNet20 section of Table 1. Is my understanding correct that the "ResNet20" results are for a pair of BNNs trained from scratch, while the "ResNet20*" results are for a pair of BNNs trained with the approximate posterior means set to the values from a deterministic model?
  - Is it correct that all results (all three metrics across all three approaches) are computed after averaging the predicted probs from the pair of models?
- For the experiments, a deterministic baseline is missing. Given the intro that discusses how BNNs can lag behind deterministic models in acc (though not always), the experiments lack a comparison. It would be helpful to understand how the proposed approach compares to a deterministic baselines, specifically a single deterministic model and a size-2 deep ensemble. Could you add this as a baseline? I would consider this to be a blocker for the paper given the framing and relevant literature.

Other:
- The KL divergence term is scaled by the square of the temperature -- why?
- How did you choose the values for temp, alpha, and beta? They differ between CIFAR-10/100 and ImageNet. Did you ablate values?

Minor comments:
- updating lines 17 & 22 of Alg 1 could be helpful for readability

**Limitations:**

No limitations are included.

---

> ### Author Rebuttal · Authors · 2023-08-09
>
> ## Weaknesses
> 1. A few key details are unclear in the text, and importantly a deterministic baseline is missing.
>
> **Answer:** We acknowledge the reviewer's comment. Accordingly, we present here the results of the deterministic baseline in the following table.  The results indicate that our BNN models that initialized with the mean value from the pre-trained deterministic models outperform the deterministic models in all metrics ACC, NLL, ECE.
>
> Table: Comparative results of our proposed method with deterministic models and 2-size deep deterministic ensemble in terms of Top-1 classification accuracy, NLL, and ECE on CIFAR-100 dataset. *Bayesian neural networks are initialized with the mean value from the pre-trained deterministic model. DNN means the deterministic model (ResNet20 or ResNet32). Size-2 DNN means deep ensemble of 2 deterministic models. For a size-2 deterministic deep ensemble, we separately train two deterministic models, which have the same architecture, but with different initializations. After training, we ensemble 2 models by averaging the probability outputs.
>
> |                |          | ACC↑            |          |          |          | NLL↓            |          |          |          | ECE↓            |          |          |
> |----------------|----------|------------------|----------|----------|----------|------------------|----------|----------|----------|------------------|----------|----------|
> |                | DNN      | size-2 DNN       | DML      | Ours     | DNN      | size-2 DNN       | DML      | Ours     | DNN      | size-2 DNN       | DML      | Ours     |
> | ResNet20       | 69.13    | 71.91            | 67.27    | 68.32    | 1.106    | 0.979            | 1.174    | 1.101    | 0.065    | 0.058            | 0.057    | 0.041    |
> | ResNet20*      | -        | -                | 69.61    | 70.45    | -        | -                | 1.073    | 1.043    | -        | -                | 0.047    | 0.038    |
> | ResNet32       | 71.36    | 74.10            | 68.59    | 70.53    | 1.074    | 0.924            | 1.169    | 1.029    | 0.080    | 0.061            | 0.087    | 0.043    |
> | ResNet32*      | -        | -                | 71.45    | 72.14    | -        | -                | 1.012    | 0.975    | -        | -                | 0.064    | 0.040    |
>
> ## Questions
> 1. Consider the ResNet20 section of Table 1. Is my understanding correct that the "ResNet20" results are for a pair of BNNs trained from scratch, while the "ResNet20*" results are for a pair of BNNs trained with the approximate posterior means set to the values from a deterministic model?
>
>  **Answer:**  As outlined in lines 262-265, we employ a pair of BNN models; one model is initiated from scratch, referred to as "ResNet20", while the other is initialized with the mean $\mu$ is from the pretrained deterministic model, denoted as "ResNet20*".  Thus, in Table 1, the results associated with "ResNet20" and "ResNet20*" are obtained by training a pair of BNNs, one initialized from scratch (ResNet20) and the other initialized (ResNet20*) with the mean $\mu$ from the pretrained deterministic model.
>
> 2. Is it correct that all results (all three metrics across all three approaches) are computed after averaging the predicted probs from the pair of models?
>
>   **Answer:** The average results in tables 1,2, and 3 in the main paper represent only the average of results from model 1 and model 2 in a pair of models. This will be clarified further in the paper.
>
> 3. It would be helpful to understand how the proposed approach compares to deterministic baselines, specifically a single deterministic model and a size-2 deep ensemble.
>
>   **Answer:** For the comparative results between the proposed approach and the deterministic model and a size-2 deterministic deep ensemble, please refer to the table above. The results show that although our method's accuracy and NLL are lower than those of the size-2 deterministic deep ensemble, our method surpasses the deep ensemble in terms of ECE.
>
> 4. The KL divergence term is scaled by the square of the temperature -- why?
>
> **Answer:** We adopt the distillation method using soft logits proposed by Hinton et al. [13]. The hyper parameter $T$ controls the smoothness level of the prediction distribution. As $T$ increases, the prediction distribution becomes more smooth.  It's worth noting that the magnitudes of the gradients, which derive from the soft targets, are inversely proportional to $T^2$. Therefore, we scale the KL divergence term by $T^2$.
>
> 5. How did you choose the temperature parameter (T) values,$\alpha$, and $\beta$? They differ between CIFAR-10/100 and ImageNet. Did you ablate values?
>
> **Answer:**  Please refer to the general response.
>
> 6. Updating lines 17 and 22 of Alg 1 could be helpful for readability
>
> **Answer:** We thank the Reviewer for your suggestion. We have updated the line 17 as follows:
>
> compute the total loss for B1:
> $\mathcal{L}^{B1} = \mathcal{L}{\text{logits}}^{B1} + \alpha \mathcal{L}{\text{diverse param}} + \beta \mathcal{L}_{\text{diverse feat}}^{B1}$
>
> and line 22 as follows:
>
> compute the total loss for B2:
> $\mathcal{L}^{B2} = \mathcal{L}{\text{logits}}^{B2} + \alpha \mathcal{L}{\text{diverse param}} + \beta \mathcal{L}_{\text{diverse feat}}^{B2}$

---

> > ### Author Response · Authors · 2023-08-17
> >
> > Dear Reviewer tB5V,
> >
> > We hope the Reviewer has had time to look at our rebuttal. Could the Reviewer please share with us the Reviewer’s feedback on it?
> >
> > We sincerely appreciate the time and effort the Reviewer has dedicated to evaluating our paper and our response.

---

> > ### Author Response · Authors · 2023-08-19
> > **Rebuttal to tB5V**
> >
> > Dear tB5V
> >
> > Could you look at the rebuttal and check if the authors addressed your concerns? Deadline is in just 2 days.
> >
> > Best regards, your AC

---

> > > ### Comment · Area_Chair_jypT · 2023-08-19
> > >
> > > @Authors: In the future, please refrain from impersonating me; the PCs and SACs might not look too kindly on this kind of behavior.
> > >
> > > @Reviewer tB5V: Could you please acknowledge to the authors that you have read their rebuttal and potentially update your score in case your concerns have been addressed?
> > >
> > > Best regards,
> > > The actual AC

---

> > > > ### Author Response · Authors · 2023-08-19
> > > > **My mistake**
> > > >
> > > > Sorry about the reply requesting the review by tB5V. I mistakenly assumed that this was one of the papers I was chairing. I had no intention of impersonating the AC. Please accept my apologies.
> > > > Best regards.

---

### Official Review · Reviewer_xPiJ · 2023-07-09

**Soundness:** 3 good
**Presentation:** 3 good
**Contribution:** 3 good
**Rating:** 5
**Confidence:** 4

**Summary:**

This paper presents a novel method for enhancing the performance of Bayesian Neural Networks (BNNs) by employing deep mutual learning. The proposed approach aims to enhance the diversity of both network parameter distributions and feature distributions, encouraging individual networks to capture unique characteristics of the input data. The effectiveness of the proposed method is demonstrated on datasets, including CIFAR10, CIFAR100, and ImageNet.

**Strengths:**

The proposed method improves performance and uncertainty estimation while reducing the expected calibration error (ECE).  The technical approach is novel as the method introduces mutual learning in the context of BNNs and first to propose maximizing the distance between feature distributions and parameter distributions. The paper includes large scale data experiments (ImageNet) and ablation studies to demonstrate the effectiveness of each technical contribution introduced in this paper.

**Weaknesses:**

The previous studies mentioned in the paper utilize alignments on feature maps [4] or predictions [38], rather than diversifying them. In contrast, the proposed method diversifies both feature distributions and parameter distributions which is an opposite approach to the previous works. Interestingly, both alignment-based and diversification methods improves performance over vanilla BNNs, as indicated in Table 1, 2, 3, and 5. However, the paper does not explicitly explain the reasons behind the performance improvements resulting from these contrasting approaches.

Given the observed contradicting results in the experiments, where the alignment-based method (DML [38]) also enhances the performance of BNNs, an important question arises: could combining alignment-based methods with parameter diversification further improve BNN performance? Alternatively, is it necessary to diversify both feature and parameter distributions to achieve significant improvements?

In the experiment section, the proposed method is only compared with [38] and not with [4].

Hyperparameters used for CIFAR experiments and ImageNet experiments are different. However, the paper does not describe details regarding the hyperparameter tuning or determination.

**Questions:**

3 block resnet is used for CIFAR experiments while 4 block resnet is used for ImageNet experiments. Why different form of resents are used for different datasets?

**Limitations:**

Limitations are shortly addressed in the supplementary.

---

> ### Author Rebuttal · Authors · 2023-08-09
>
> ## Weaknesses
>
> 1. The previous studies mentioned in the paper utilize alignments on feature maps [4] or predictions [38], rather than diversifying them. In contrast, the proposed method diversifies both feature distributions and parameter distributions which is an opposite approach to the previous works. Interestingly, both alignment-based and diversification methods improves performance over vanilla BNNs, as indicated in Tables 1, 2, 3, and 5. However, the paper does not explicitly explain the reasons behind the performance improvements resulting from these contrasting approaches.
>
> **Answer:**
> Our method does not contradict to the original DML work [38]. Actually, both ours and [4] are built on top of [38]. Specifically, consider a pair of peer networks, to encourage each network to learn and teach each other, DML [38] tries to match the prediction distributions of the two networks through minimizing the KL distance. Following [38], both ours and [4] have the KL loss term for each network loss.
> Comparing to [4], ours and [4] are two different approaches for enhancing the mutual learning. The idea of [4] is that for each network, in addition to learning useful features for the  task on its own, [4] also encourages each network to learn the feature distribution of its peer network through an adversarial training strategy. By doing this, they encourage both networks to learn features that generalize better. In the other words, they encourage the two networks to learn features that work well for both.  Different from [4], our approach encourages each network to learn a different set of features that are good for the task. In the other words, our approach encourages each peer network to capture different characteristics of the input. It is worth noting that our approach encourages the diversity in intermediate features. For the last layer, we have the soft logit alignment similar to DML to encourage each network to learn and teach from each other.
> To summarize, ours and [4] are two different approaches to enhance mutual learning. [4] aims to learn generalized features, while ours aims to make each network to learn different characteristics of the input.
>
> Combining our idea and [4] may further improve mutual learning, however, this is not the focus of our paper. It is also worth noting that in [4] the authors do not focus on Bayesian Neural Networks. In addition, their implementation is not publicly available. This makes it difficult for us to compare to [4] in the context of BNNs. In the efforts to compare to [4], we reimplement the approach in [4] for the BNNs context. To make a fair comparison, both ours and [4] use ResNet20 as the backbone. For the discriminator of [4], we follow the description in their paper. The comparisons between ours and [4] on CIFAR100 are in the following table. The results show that both ours and [4] improve the performance of BNNs. However, our approach achieves better performance than [4] in terms of ACC, NLL, and ECE.
>
> Table: Comparative results of our proposed method with deterministic models and [4] in terms of Top-1 classification accuracy, NLL, and ECE on CIFAR-100 dataset.*Bayesian neural networks are initialized with the mean value from the pre-trained deterministic model. DNN means the deterministic model.
>
> |             | ACC↑       |        |       |     | NLL↓       |       |       |       | ECE↓       |        |      |      |
> |-------------|-------------|-------------|-------------|-------------|-------------|-------------|-------------|-------------|-------------|-------------|-------------|-------------|
> |             | DNN         | DML         | [4]         | Ours        | DNN         | DML         | [4]         | Ours        | DNN         | DML         | [4]         | Ours        |
> | ResNet20    | 69.13       | 67.27       | 68.18       | 68.32       | 1.106       | 1.174       | 1.154       | 1.101       | 0.065       | 0.057       | 0.047       | 0.041       |
> | ResNet20*   | -           | 69.61       | 70.26       | 70.45       | -           | 1.073       | 1.039       | 1.043       | -           | 0.047       | 0.045       | 0.038       |
>
> 2. The paper does not describe details regarding the hyperparameter tuning or determination.
>
> **Answer:** Please refer to the general response.
> ## Questions
> 1. 3 block resnet is used for CIFAR experiments while 4 block resnet is used for ImageNet experiments. Why different form of resents are used for different datasets?
>
> **Answer:**  Regarding the model architectures when evaluating on CIFAR100 and ImageNet, we follow the previous works in knowledge distillation [4, 12, 13, 24, 38, 39].
> Typically, for large datasets like ImageNet with an input size of $224\times 224$, people often use ResNet models with 4 blocks such as ResNet18. Meanwhile, for CIFAR-10 and CIFAR-100, which have smaller input size, i.e., $32 \times 32$, people usually utilize ResNet models with 3 blocks such as ResNet20, ResNet32, and ResNet56. These models have much fewer channels per convolutional layer and a reduced number of blocks (3 instead of 4).
>   It is worth noting that although the ResNet20, ResNet32, and ResNet56 models (3 blocks) are deeper than ResNet18 (4 blocks), they have significantly fewer channels per layer compared to ResNet18. For example, the last conv. layer of ResNet20 has 64 output channels, while the last conv. layer of ResNet18 has 512 output channels.

---

> > ### Author Response · Authors · 2023-08-17
> >
> > Dear Reviewer xPiJ,
> >
> > We hope the Reviewer has had time to look at our rebuttal. Could the Reviewer please share with us the Reviewer’s feedback on it?
> >
> > We sincerely appreciate the time and effort the Reviewer has dedicated to evaluating our paper and our response.

---

> > ### Comment · Reviewer_xPiJ · 2023-08-18
> > **Thank you for the response.**
> >
> > Thank you for clarifying my questions.
> > Based on the reviews and responses, I raise my score to above borderline.

---

> > > ### Author Response · Authors · 2023-08-18
> > >
> > > Dear Reviewer xPiJ,
> > >
> > > We thank the Reviewer for increasing the score. We greatly appreciate the time and effort the Reviewer dedicated to considering our paper and our response.

---

### Author Rebuttal · Authors · 2023-08-09

We sincerely thank the reviewers for the constructive feedback.

## General response to the choice of hyper parameters
Regarding hyper parameters $T$, $\alpha$, $\beta$:
For the temperature $T$, we follow the seminal work [13] and [4].  This parameter $T$ controls the smoothness of the prediction distribution. As the value of $T$ increases, the prediction distribution becomes smoother. The hyper parameters $\alpha$ and $\beta$ control the impacts of the diversity of network parameter distributions and network feature distributions on the learning of a pair of peer networks. We present here the ablation studies on the choice of $T$, $\alpha$, $\beta$ on CIFAR100. We denote "*" mean Bayesian neural networks are initialized with the mean value from the pre-trained deterministic model.

For ablation studies for parameter T, we vary the value of $T$ from 1 to 5, and fix the values of $\alpha=1$ and $\beta=2$. The results are shown in the following table. The results show that the best value of $T$ is 3.
|  T   | 1     | 2     | 3     | 4     | 5     |
|---------|-------|-------|-------|-------|-------|
| ResNet20| 66.6  | 67.66 | 68.32 | 67.95 | 67.93 |
| ResNet20* | 69.62 | 70.14 | 70.45 | 70.12 | 69.97 |

For ablation studies for parameter $\alpha$, we vary $\alpha$ when promoting diversity in parameter space, and set the value of $T=3$ and $\beta=0$. The results are shown in the following table. The results show that by setting $\alpha = 1$, we achieve a better performance compared to other tested values of $\alpha$.
| $\alpha$ | 0.1   | 1     | 2     | 5     | 10    |
|--------------|-------|-------|-------|-------|-------|
| ResNet20     | 67.25 | 67.78 | 67.18 | 67.45 | 67.29 |
| ResNet20*    | 69.79 | 70.22 | 69.92 | 69.61 | 69.48 |

For ablation studies for parameter $\beta$, we vary $\beta$ when promoting diversity in feature space, and set the value of $T=3$ and $\alpha=0$. The results are shown in the following table. The results show that by setting $\beta = 2$, we achieve a better performance compared to other tested values of $\beta$.
| $\beta$  | 0.1   | 1     | 2     | 5     | 10    |
|--------------|-------|-------|-------|-------|-------|
| ResNet20     | 66.95 | 67.17 | 67.57 | 67.14 | 67.09 |
| ResNet20*    | 69.84 | 69.91 | 70.04 | 69.85 | 69.90 |

In addition, from our experiments, we found that the accuracies are only  slightly different when $\alpha$ and $\beta$ take values in {1, 2}.

---

### Decision · Program_Chairs · 2023-09-21

**Decision:**

Accept (poster)

**Comment:**

The reviewers praise the novelty and originality of the method, the strong empirical results, the comprehensive experiments and the writing of the paper. The main criticisms are the choice of hyperparameters, the connection between this mutual learning with diversification vs existing approaches to mutual learning encouraging alignment, lack of comparison to a deterministic model, limited scope of the experiments in terms of different models, lack of evaluation on OOD data, and lack of comparison to a MCMC baseline. However, most of these points have been addressed in the rebuttal.